

# An interpretable multi-transformer ensemble for text-based movie genre classification

Faheem Shaukat[1], Naveed Ejaz[2], Zeeshan Ashraf[3], Mrim M. Alnfiai[4], Nouf Nawar Alotaibi[5] and Salma Mohsen M. Alnefaie[6]

[1] Department of Computing and Technology, IQRA University, Islamabad, Pakistan
[2] School of Computing, Queen's University, Kingston, Canada
[3] Department of Computer Science, IISAT, Gujranwala, Pakistan
[4] Department of Information Technology, College of Computers and Information Technology, Taif University, Taif, Saudi Arabia
[5] Department of Special Education, College of Education, Najran University, Najran, Saudi Arabia
[6] Physics Department, College of Science, Taif University, Taif, Saudi Arabia

## ABSTRACT

Multi-label movie genre classification is challenging due to the inherent ambiguity and overlap between different genres. Most of the existing works in genre classification use audio-visual modalities. The potential of text-based modalities in movie genre classification is still underexplored. This paper proposes an ensemble deep-learning model that uses movie plots to predict movie genres. After pre-processing the text plots, three transformer-based models, Bidirectional Encoder Representations from Transformers (BERT), DistilBERT, and Robustly Optimized BERT Pre-training Approach (ROBERTa), are used to generate genre predictions, combined through a weighted soft-voting method. The proposed ensemble architecture achieves state-of-the-art performance on two benchmark datasets, Trailers12K and LMTD9, with a micro-average precision of 80.10% and 80.37%, respectively, significantly outperforming both traditional machine learning approaches and advanced deep learning models. The ensemble's superior performance is attributed to its ability to combine the diverse strengths of individual models and capture nuanced genre-specific information from textual features. The lack of interpretability in deep learning models for genre classification is addressed using Local Interpretable Model-Agnostic Explanations (LIME), which provides both local and global explanations for the model's predictions. The findings of the study highlight the potential of textual data in automated genre classification and emphasize the importance of interpretability methods in multi-label genre classification.

Corresponding author
Zeeshan Ashraf,
zeeshan.ashraf@ieee.org,
zeeshan.np@gmail.com

# INTRODUCTION

Movie genre classification is one of the important problems in multimedia content analysis and recommendation systems (*Deldjoo et al., 2020*). In multi-label genre classification, movies may belong to multiple genres simultaneously, making it a complex multilabel classification problem (*Shaukat et al., 2024*; *Wehrmann & Barros, 2017*). For instance, a

movie can be categorized as both "action" and "comedy" or "drama" and "romance". The complexity of the problem demands sophisticated approaches with an ability to capture intricate genre relationships. The exponential growth of digital movie content across streaming platforms and online databases has made automated genre classification increasingly important for content organization (*Mehal et al., 2021*), recommendation systems (*Radhika & Swaraj, 2020*), and market analysis (*Chen, 2021*).

The production houses usually assign one or more genres to movies during their release. However, automated genre classification is still a research challenge for many reasons (*Kumar et al., 2023*; *Vahed, Tabatabaei & Taherkhani, 2024*). The genre assignments often vary across different platforms and regions. For example, a film categorized as a "thriller" on one platform might be labelled as a "mystery" or "crime" on another. Also, the sharp rise in user-generated content, independent films, and international cinema has resulted in media content that often does not have clear genre labels. Moreover, genre definitions shift over time; therefore, flexible classification systems are needed to keep up with these changes (*Davids, 2023*). Automated genre classification can also offer more detailed and nuanced predictions than a manual assignment of genres (*Alsekait et al., 2024*). Lastly, automatic genre classification enables retrospective analysis, which is valuable for film studies research and improvement in content recommendation and discovery (*Česálková, 2024*).

Text-based multilabel genre classification, using modalities like plot summaries (*Ertugrul & Karagoz, 2018*; *Kim et al., 2024*; *Matthews & Glitre, 2021*), synopses (*Wang, 2020*), and reviews (*González et al., 2023*), is an emerging area in multi-label genre classification. Traditionally, most methods have focused on audiovisual features from modalities like posters (*Narawade et al., 2021*; *Kundalia, Patel & Shah, 2020*) and trailers (*Shaukat et al., 2024*; *Sharma et al., 2021*; *Shambharkar et al., 2020*). However, the textual content of movie descriptions often contains subtle linguistic cues, thematic elements, and narrative patterns that can significantly enhance genre prediction accuracy. Moreover, text-based approaches offer practical advantages in terms of computational efficiency and data accessibility compared to video-based methods (*Zhang et al., 2024*).

Additionally, a significant limitation of the existing genre classification approaches is their "black box" nature, making it difficult to comprehend the reasoning behind genre predictions. More interpretable approaches that provide clear explanations for genre classifications and identify specific textual or narrative elements that influence genre predictions are needed. This enhanced interpretability would improve trust in the classification systems and provide valuable insights for film studies and content creation.

This study examines the effectiveness of utilizing various machine learning and transformer-based architecture variants for movie genre classification based on movie plots. An ensemble pipeline utilizing three different variants of transformer-based models, namely Bidirectional Encoder Representations from Transformers (BERT) (*Devlin et al., 2018*), Robustly Optimized BERT Pre-training Approach (RoBERTa) (*Liu et al., 2019*), and Distilled BERT (DistilBERT) (*Sanh et al., 2019*) has been proposed for predicting movie genres. A thorough explainability analysis using Local Interpretable Model-Agnostic

Explanations (LIME) (*Ribeiro, Singh & Guestrin, 2016*) has been conducted from both local and global viewpoints to clarify the decision-making process of the proposed method.

Today's streaming platforms like Netflix, Amazon Prime, and Disney+ use intelligent systems to recommend shows and movies. Since many movies span multiple genres, accurate multi-label classification helps suggest content users are more likely to enjoy and improves library organization. *Alsekait et al. (2024)* shows that genre tags enhance recommendations and help users find content more easily.

The main goals of this study are: (1) to develop a robust text-based multi-label genre classification model using transformer architectures, (2) to show the effectiveness of early genre prediction using plot summaries alone, and (3) to enhance model interpretability through LIME for both local and global explanations. This approach is particularly useful when visual or audio content is unavailable, enabling earlier and more efficient genre tagging.

The rest of the paper is organized as follows: "Related Work" provides a comprehensive review of the relevant literature, while "Methodology" introduces the proposed methodology, including the explainability mechanism. "Experiments and Results" presents the experimental setup, including the datasets, evaluation metrics, and the results obtained. Finally, "Conclusions" concludes the paper with a summary of the key findings and potential avenues for future research.

# RELATED WORK

Researchers have employed various modalities for multi-label genre classification, including trailers (*Montalvo-Lezama, Montalvo-Lezama & Fuentes-Pineda, 2023*), posters (*Popat et al., 2023*), audio (*Sharma et al., 2021*), textual data (*Rajput & Grover, 2022*), and a combination of these modalities (multi-modal methods) (*Cai et al., 2023*). This paper only focuses on a literature review of textual methods.

## Synopsis-based approaches

In recent years, movie synopses have become valuable for multi-label genre classification, providing a concise overview of a film's plot. Several studies have investigated approaches to using these synopses for automated classification. For instance, *Portolese & Feltrin (2018)* and *Kim et al. (2024)* concentrated on extracting text-based features such as term frequency-inverse document frequency (TF-IDF) and word embeddings, finding that models like logistic regression (LR) and multi-layer perceptron (MLP) with TF-IDF features outperformed other methods. Similarly, *Akbar, Utami & Yaqin (2022)* and *Buslim et al. (2022)* conducted comparative studies on ML algorithms, including support vector machines (SVM), LR, and Naive Bayes (NB), consistently identifying SVM as the top performer in their experiments. Expanding on this, *Wang (2020)* explored both ML and deep learning (DL) methods, concluding that recurrent neural networks (RNNs) with long short-term memory (LSTM) layers achieved the highest accuracy in genre classification when applied to online movie synopses.

## Subtitle-based approach

Subtitles are textual versions of dialogue and sound effects in movies. They are often used for accessibility and translation. In genre classification, subtitles can be a rich source of linguistic data, capturing a movie's script's tone, language, and context.

_Rajput & Grover (2022)_ used movie subtitles for multi-label genre classification by identifying high-frequency words associated with specific genres to train machine learning models. _Hasan et al. (2021)_ proposed a model using a parameter-optimized hybrid classifier (POHC), which combined SVM and Decision Trees (DT). The authors used a dataset of 1,000 movie subtitle files, and their results indicated that the hybrid POHC approach outperformed traditional machine learning classifiers.

## Movie plots

Movie plots are a condensed summary of the storyline. _Ertugrul & Karagoz (2018)_ used a deep learning approach with bidirectional LSTM networks to classify movie genres based on plot summaries. They improved their model's performance by dividing the plot summaries into sentences and training the network on these segments. Their model outperformed traditional RNN and logistic regression models. _Kumar et al. (2023)_ introduced a method that merged problem transformation techniques, like binary relevance and label powerset, with different text vectorizers, including a count vectorizer and TF-IDF. _Matthews & Glitre (2021)_ applied unsupervised topic modelling to textual movie plots. They utilized the model's topic proportions to investigate genre characteristics, temporal shifts, and canonicity. _Kumar et al. (2021)_ used the universal sentence encoder (USE) for text encoding, combining it with a sequential neural network model and label powerset for problem transformation using movie plots.

## Research gaps

Unlike multimodal and audio-visual models such as TimeSformer (_Bertasius, Wang & Torresani, 2021_) and DIViTA (_Montalvo-Lezama, Montalvo-Lezama & Fuentes-Pineda, 2023_), which rely on trailers or video data requiring high computational resources and are sensitive to visual noise, text-based models are lightweight, easier to deploy, and provide clearer semantic cues. Visual methods also tend to lack interpretability due to the opaque nature of visual features. Additionally, resources like trailers, posters, or audio clips are often released late in the production cycle, limiting early-stage genre prediction. In contrast, plot summaries and scripts are typically available earlier, making text-based approaches more practical for early genre classification, which aids in timely decisions for marketing and content recommendation.

Most existing studies using textual modalities focus on traditional ML algorithms (_e.g._, LR, SVM, DT) and DL models such as RNNs and LSTMs. Although these models have shown some success, recent advancements in Transformer-based models have not been widely explored for movie genre classification. Studies such as _Ertugrul & Karagoz (2018)_ and _Wang (2020)_ have applied older architectures, including LSTM and bidirectional LSTM, which are less effective in capturing long-range dependencies in text than transformers.

**Table 1  Comparative summary of text-based genre classification methods.**

| Studies | Text source | Methodology | Contributions/Limitations |
|---|---|---|---|
| *Portolese & Feltrin (2018)* | Synopsis | TF-IDF + Logistic Regression | Traditional ML baseline, limited contextual understanding |
| *Ertugrul & Karagoz (2018)* | Plot summaries | BiLSTM | Captures sequences but struggles with long dependencies |
| *Wang (2020)* | Online synopses | RNN-LSTM | Temporal modeling with slow convergence |
| *Kumar et al. (2021)* | Movie plots | Universal Sentence Encoder | Improved embeddings but no ensemble learning |
| *Rajput & Grover (2022)* | Subtitles | Hybrid SVM-DT classifier | Focused on dialogue features only |
| Proposed method | Plot | BERT, RoBERTa, DistilBERT + LIME | First transformer ensemble for genre classification and interpretability with LIME |

Although some studies used hybrid models (*e.g.*, hybrid SVM-Decision Tree model (*Hasan et al., 2021*)), there is a limited exploration of ensemble techniques involving multiple state-of-the-art models, there is potential to improve classification accuracy by combining different models that capture diverse aspects of the text data.

Several studies used traditional text embedding techniques such as TF-IDF and word embeddings. While *Kumar et al. (2021)* used USE, few studies explore more recent and powerful pre-trained language models such as BERT, which can offer richer contextualized embeddings. This indicates a gap in adopting modern text encoding techniques for more accurate genre classification.

Moreover, to the best of our knowledge, none of the genre classification methods in the literature have employed explainability and interoperability techniques to clarify the classification process. To further clarify the differences between our work and previous studies, Table 1 presents a comparative summary of related approaches focused on text-based movie genre classification.

## METHODOLOGY

This study explores a multi-label classification task aimed at predicting movie genres by utilizing an ensemble of transformer-based models applied to movie plots. Movie plots provide a concise storyline summary, highlighting key characters, events, themes, and conflicts, making them valuable textual data for identifying genres. Each data sample comprises a plot synopsis paired with one or more associated genre labels, reflecting the multi-label nature of the task.

Consider $D = \{(x_1, y_1), (x_2, y_2), \ldots, (x_n, y_n)\}$ as a dataset of movie plot synopses and their corresponding genre labels. $x_i$ represents the textual plot synopsis of the $i$-th movie. $y_i \subseteq Y$ is the set of genre labels associated with the $i$-th movie, where $Y = \{g_1, g_2, \ldots, g_m\}$ is the set of all possible genre labels (*e.g.*, "Action," "Comedy," "Drama," *etc.*). The task is to learn a function $f : X \rightarrow 2^Y$ that maps a movie plot synopsis $x \in X$ to a set of predicted genre

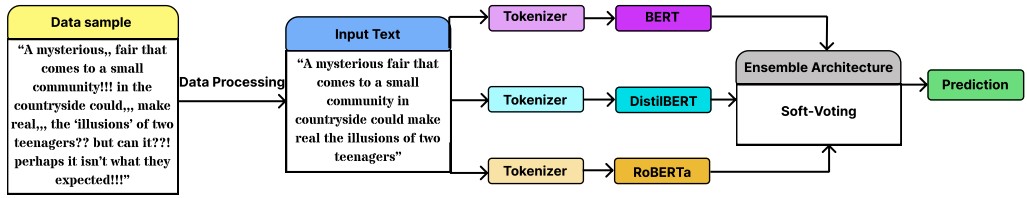

**Figure 1   Overview of the proposed pipeline.**

labels $\hat{y} \subseteq Y$. This function aims to identify the most relevant subset of genres from the full label set $Y$ for a given input text $x$.

Figure 1 illustrates an overview of the proposed pipeline. Initially, the input text was subjected to data preprocessing before being tokenized. Three separate models then process the tokenized data. The final genre category of the input text was determined by combining their outputs through soft voting. As shown in Fig. 1, the pipeline ensures that each model processes the same tokenized input independently, enabling diverse yet complementary perspectives on genre prediction.

In ML models, TF-IDF was employed to convert movie plots into numerical vectors, enabling effective feature extraction.

## Data preprocessing

Several pre-processing steps were applied to clean and standardize the text. Punctuation that did not contribute to the meaning of the plot, such as commas and periods, was removed, while elements that could affect sentiment or structure, like question marks, were kept. Special characters (*e.g.*, @, #), which were irrelevant to movie plots, were also removed.

Stop word reduction was performed by minimizing commonly used words (*e.g.*, "the," "is") without completely eliminating them, as some still carry contextual significance. Contractions (*e.g.*, "didn't") were expanded to maintain consistency and preserve subtle meanings. Redundant phrases or terms were filtered out to avoid inflating the text without adding value. Spacing inconsistencies were corrected for uniformity. These steps helped create a clean dataset suitable for further processing. Table 2 presents an example of pre-processing applied to a sample movie plot.

These preprocessing steps ensure that the textual data fed into transformer models is semantically meaningful, syntactically consistent, and devoid of irrelevant noise, thereby improving the model's ability to extract genre-relevant features.

## Classifier networks

Transfer learning was used by leveraging three classifier networks pre-trained on generalized tasks across different domains. Recent research indicates that choosing between two to five independent, diverse models strikes the best balance between training time and prediction accuracy (*Mohammed & Kora, 2023*). Three models are selected to ensure a reasonable training duration and to avoid ties during model voting.

**Table 2   Step-by-step transformation of a movie plot after applying pre-processing steps.**

| Preprocessing step | Movie plot |
| --- | --- |
| **Original plot** | John, a brilliant scientist, embarks on a dangerous journey to stop an evil organization. He can't do it alone—he teams up with Sarah, a skilled hacker. Together, they must infiltrate the organization's Headquarter, which is heavily guarded. But will they make it in time? It's a race against the clock! |
| **Punctuation cleaning & special character removal** | John a brilliant scientist embarks on a dangerous journey to stop an evil organization He cant do it alone he teams up with Sarah a skilled hacker Together they must infiltrate the organizations Headquarter which is heavily guarded But will they make it in time Its a race against the clock |
| **Contraction expansion** | John a brilliant scientist embarks on a dangerous journey to stop an evil organization He cannot do it alone he teams up with Sarah a skilled hacker Together they must infiltrate the organizations Headquarter which is heavily guarded But will they make it in time It is a race against the clock |
| **Stop word reduction** | John brilliant scientist embarks dangerous journey stop evil organization cannot alone teams Sarah skilled hacker Together infiltrate organizations Headquarter heavily guarded race against clock |
| **Redundancy removal, spacing consistency, & sentence structure consistency** | John brilliant scientist embarks dangerous journey stop evil organization cannot alone teams Sarah skilled hacker infiltrate organizations Headquarter heavily guarded race against clock |

In this study, four models were experimented with, including BERT (*Devlin et al., 2018*), RoBERTa (*Liu et al., 2019*), AlBERT (*Lan et al., 2019*), and DistilBERT (*Sanh et al., 2019*). These models are based on Transformer architecture (*Vaswani et al., 2017*). These models utilize a bidirectional architecture, allowing for examining input text from both the left and right sides.

BERT (*Devlin et al., 2018*) is a DL model for pre-training contextualized word representations. Its bidirectional nature makes it very important to handle the nuanced meanings of words in context. RoBERTa (*Liu et al., 2019*) is an optimized variant of BERT that improves performance by training on much larger datasets, totalling 160 GB, including CC-News (76 GB), OpenWebText (38 GB), and Stories (31 GB). Additionally, RoBERTa uses a dynamic masking strategy during masked language model (MLM) training, generating new masking patterns for each epoch to enhance generalization. DistilBERT (*Sanh et al., 2019*) is a smaller, faster version of BERT designed to reduce the model size and inference time while retaining much of BERT's language understanding. It uses knowledge distillation, transferring knowledge from a larger model (teacher) to a smaller one (student), with a distillation loss function combining cross-entropy loss and Kullback–Leibler (KL) divergence. DistilBERT reduces the number of transformer layers from 12 to 6, retains the same hidden size (768) and attention heads (12), and omits the token-type embeddings and next sentence prediction (NSP) objective. It uses the MLM objective during pre-training, similar to BERT. AlBERT (*Lan et al., 2019*) is a derivative of

the BERT model designed to address the challenges of large-scale pre-training regarding both memory consumption and training time.

The fine-tuning process for the BERT, AlBERT, RoBERTa, and DistilBERT models on the Trailers12K dataset for movie genre classification was carefully designed to adapt these models for the multi-label classification task. After obtaining results from these four models, ALBERT was excluded due to its lower performance compared to the other models. These model selections are also based on their proven effectiveness in NLP tasks and their complementary strengths.

## Fine-tuning details
Each model was modified to suit the multi-label classification task. The following changes were made:

### Output layer modification
The output layer of each model was adapted for multi-label classification. Unlike single-label classification, which uses a softmax activation function to predict one label among many, multi-label classification requires predicting multiple independent labels. To enable this, the softmax function was replaced with a sigmoid activation function for each output node. The sigmoid function outputs a value between 0 and 1, indicating the probability of each genre label being present.

### Training details
Each model—BERT, AlBERT, RoBERTa, and DistilBERT—was fine-tuned over three epochs. This duration provided sufficient time for learning while reducing the risk of overfitting. A consistent fine-tuning process was used across all models to ensure fair comparison based on architectural and pre-training differences rather than tuning variations. During training, a maximum sequence length of 256, a batch size of 32, and a learning rate of $2 \times 10^{-5}$ were used. To further prevent overfitting, a dropout rate of 0.1 was applied. The binary cross-entropy loss function and the AdamW optimizer were employed for training.

## Ensemble of fine-tuned deep learning models
We experimented with majority voting (hard voting) and probability averaging (soft voting) to create an ensemble for genre classification. In majority voting, predictions from all three models were combined through a hard voting mechanism, where each model's output contributed equally to the final decision. In probability averaging, each model outputs a probability distribution over the possible genres for each movie, representing the model's confidence in assigning each genre label, which is then used to combine the predictions. Soft voting aggregates these probability distributions by averaging them across the models. Let $P_{i,j}$ denote the probability predicted by the $i$-th model for the $j$-th genre. The ensemble probability for genre $j$, denoted as $P_j$, is computed as the mean of the predicted probabilities from all $n$ models, as shown in the following equation:

$$P_j = \frac{1}{n}\sum_{i=1}^{n} P_{i,j}. \tag{1}$$

In our case, $n = 3$, representing the BERT, RoBERTa, and DistilBERT models. The final genre labels are determined by applying a threshold $\tau$ to these averaged probabilities. A genre is assigned if its corresponding $P_j$ exceeds $\tau$:

$$\hat{y}_j = \begin{cases} 1 & \text{if } P_j \geq \tau \\ 0 & \text{if } P_j < \tau \end{cases}.$$

(2)

The experiments indicated that soft voting yielded better performance as it enabled all the models in the ensemble to contribute equally to the final prediction. This method allows each model's confidence to influence the final prediction, rather than relying solely on discrete labels, making it more suitable for fine decisions in multi-label settings.

## Interpretability for ensemble architectures

In the proposed multi-label genre classification model, LIME was used to improve the interpretability and transparency of our ensemble's predictions. Since the ensemble model functions like a black box, it was difficult to understand how specific input features influenced its predictions. LIME helps generate interpretable explanations, showing why certain genres were predicted for a given movie.

This is particularly important in multi-label classification, where it's essential to understand how features like plot keywords and linguistic patterns relate to predicted genres. LIME works by perturbing input data and observing how small changes affect predictions, highlighting which features (*e.g.*, key plot elements or themes) contributed to each genre label. Since genres often share overlapping features, LIME clarifies the unique contribution of each feature to multiple labels, offering a detailed breakdown of the model's decision-making.

The use of LIME improved the trustworthiness and accountability of the model. This is crucial in applications like movie recommendations, where stakeholders must trust the model's predictions. LIME enables validation of these predictions, ensuring they align with human intuition or reveal areas needing improvement.

### *Local explanation*

The goal of local explanation is to explain the decisions made by the ensemble model when presented with an input instance $x$ and to produce predictions $f(x)$ for various genres linked to that instance. To achieve a unified explanation, we utilize a more interpretable linear regressor. This linear regression model aims to capture the behaviour of complex black-box models by concentrating specifically on the instance being analyzed. The proposed explanation method applies Word-level perturbation to the input instance $x$. In perturbation, the words are randomly substituted with their synonyms, added or removed, word positions are rearranged, specific words are masked or omitted, and the perturbation of n-grams. This step is performed to capture local linguistic patterns. Five thousand unique perturbed samples are generated from the input data to create local explanations by randomly altering specific features (*e.g.*, removing or changing plot keywords). This comprehensive perturbation procedure created a dataset of perturbed instances surrounding the original input. The black-box ensemble model is then used to

evaluate these perturbed instances to determine the class-wise probabilities for multiple genres.

If the prediction for movie plot sample $x_i$ from the $k$-th transformer-based classifier is given by $p_i = \hat{y}_k \subseteq Y$, where $i \in [1, 5000]$ and $k \in [1, 3]$, the linear regressor is trained using the perturbed features, with the class-wise probabilities serving as ground truth. The coefficients of the linear regression model reflect the significance of each feature in influencing the local decision-making outcomes. To derive the importance of the ensemble feature for the input instance $x$, the regressor is trained for each transformer $k$ (from 1 to 3), modifying the class-wise probabilities in each iteration. In the end, the mean of the coefficients is computed to find the overall significance of each feature for the input instance $x$.

In this way, LIME produced a human-readable explanation detailing why the model predicted specific genres for the movie. For example, the explanation might state, "The movie is predicted to be a drama due to the presence of keywords such as 'family conflict' and 'emotional struggles,' while it is classified as a comedy because of humorous elements like 'light-hearted banter' and 'funny situations.'"

### Global explnation

For the global explanation of our ensemble model in the multi-label genre classification context, we analyze the overall feature importance across all predictions made by the $k$-th transformer-based classifiers. We define the global predictions for a given input instance $x$ as $\hat{Y} = \{\hat{y}_1, \hat{y}_2, \hat{y}_3\} \subseteq Y$, where each $\hat{y}_k$ corresponds to the predicted genre labels from the $k$-th transformer.

The coefficients from the linear regressors trained for each transformer were aggregated to compute the importance of the global feature. Let $\beta_{j,k}$ denote the coefficient associated with feature $j$ from the $k$-th transformer. The global importance score $\beta_j^{\text{global}}$ for each feature $j$ is then computed as:

$$\beta_j^{\text{global}} = \frac{1}{K} \sum_{k=1}^{K} \beta_{j,k}$$

where $K$ is the total number of transformers (in this case, $K = 3$).

## EXPERIMENTS AND RESULTS

This section provides a detailed analysis of the experiments and results used to evaluate the proposed ensemble method. It also discusses the results of local and global explainability on movie genre classification using the proposed method.

### Experimental setup

All the experiments were done on Kaggle Notebooks. it is a cloud platform that gives you a ready-to-use setup for machine learning work. The hardware had an NVIDIA Tesla P100 GPU. It came with 16 GB of VRAM and 3,584 CUDA cores. This helped with fast and smooth deep learning tasks using parallel processing. The system also had a single-core Intel Xeon CPU. It ran at 2.2 GHz and had two threads and a 56 MB cache. The machine

had 15.26 GB of RAM. It also provided 155 GB of storage to run code and manage data. Kaggle's system already includes many popular machine learning tools. These include Pandas, Tensorflow, Scikit-learn, Numpy, and Pytorch. Due to these libraries, we could effortlessly train, and test our models. This structure made it simple to run our training several times. It also aided us in obtaining the uniform results each time, which is essential for consistent experiments

## Datasets used

Two datasets are utilized in this study: Trailers12k (available at https://richardtml.github.io/trailers12k/) (*Montalvo-Lezama, Montalvo-Lezama & Fuentes-Pineda, 2023*) and LMTD-9 (available at https://github.com/jwehrmann/lmtd) (*Wehrmann & Barros, 2017*).

The Trailers12k dataset has twelve thousand movies. It comprises several types of content, like posters (images), trailers (videos), and IMDb material such as plot summaries. Each movie is multi-tagged with up to 10 genres from a set list. The distribution count of movies in each genre is not well-adjusted. Drama is a highly popular genre. It occurs in more than fifty-nine hundred movies among training, validation, and test sets. Some genres, like Fantasy and Sci-Fi, have fewer than 1,600 movies each. The dataset is split into three parts: 70% for training, 10% for validation, and 20% for testing. These splits are stratified, which means the genre distribution stays the same across all parts. Because of these features, the Trailers12k dataset is a good choice for testing multi-label classification models. It is especially useful when dealing with unbalanced genre data.

The LMTD-9 dataset contains approximately 4,007 movies, each labeled with one to three genres drawn from a set of nine. Unlike Trailers12k, it focuses solely on textual metadata, primarily plot summaries. The dataset also exhibits genre imbalance—for example, Drama and Comedy are the most frequent genres, with over 1,900 and 1,500 instances respectively, whereas Sci-Fi appears in only around 310 samples (229 train, 56 test, 25 validation). The dataset is split into training, validation, and test subsets, with at least 10% representation for each genre in every split.

In this work, model training was conducted using the three predefined training splits from Trailers12k, while LMTD-9 served as an additional benchmark to evaluate model generalizability and performance on a smaller, text-only dataset with a different genre distribution.

## Evaluation metrics

For evaluation, we employed four commonly used metrics in multi-label classification: micro average precision ($\mu$AP), macro average precision (mAP), weighted average precision (wAP), and sample average precision (sAP). $\mu$AP computes precision globally by aggregating true positives, false positives, and false negatives across all classes, treating every prediction equally. This makes it particularly effective in reflecting overall model performance regardless of class imbalance. In contrast, mAP calculates the precision for each label independently and then averages these values, giving equal importance to all labels, including those that are underrepresented. wAP addresses class imbalance by weighting each label's precision according to its frequency in the dataset, thus reflecting

**Table 3** Comparison of results with (mean ± standard deviation) and computational performance for different traditional ML and DL algorithms across various splits on the Trailers12K dataset.

| Model | µAP | mAP | wAP | sAP | Train (s) | Test (s) |
|---|---|---|---|---|---|---|
| Decision trees | 65.91 ± 0.30 | 63.84 ± 0.28 | 62.92 ± 0.28 | 63.84 ± 0.28 | 15.80 | 1.02 |
| k-nearest neighbors | 59.92 ± 0.06 | 58.41 ± 0.10 | 58.17 ± 0.18 | 58.41 ± 0.10 | 13.03 | 1.69 |
| Logistic regression | 72.66 ± 0.03 | 68.87 ± 0.07 | 70.29 ± 0.05 | 68.87 ± 0.07 | 12.27 | 1.01 |
| Naive bayes | 68.26 ± 0.09 | 63.48 ± 0.25 | 65.83 ± 0.06 | 63.48 ± 0.25 | 10.08 | 1.01 |
| Random forest | 67.19 ± 0.19 | 63.37 ± 0.20 | 64.84 ± 0.29 | 63.37 ± 0.20 | 12.29 | 1.13 |
| Support vector machine | 74.00 ± 0.04 | 71.98 ± 0.10 | 72.41 ± 0.05 | 71.98 ± 0.10 | 16.49 | 2.89 |
| Ensemble (ML Hard) | 72.12 ± 0.28 | 68.49 ± 0.24 | 69.73 ± 0.05 | 68.49 ± 0.24 | 167.61 | 75.98 |
| Ensemble (ML Soft) | 73.13 ± 0.29 | 69.50 ± 0.25 | 70.74 ± 0.06 | 69.50 ± 0.25 | 185.57 | 83.58 |
| AlBERT | 78.04 ± 0.40 | 75.51 ± 0.29 | 75.72 ± 0.29 | 75.51 ± 0.29 | 200 | 6 |
| BERT | 79.48 ± 0.39 | 78.04 ± 0.41 | 77.88 ± 0.31 | 78.04 ± 0.41 | 231 | 7 |
| DistilBERT | 79.64 ± 0.40 | 77.35 ± 0.12 | 77.34 ± 0.17 | 77.35 ± 0.12 | 180 | 5 |
| RoBERTa | 79.93 ± 0.39 | 77.58 ± 0.66 | 77.57 ± 0.48 | 77.58 ± 0.66 | 250 | 8 |
| Ensemble (DL Hard) | 72.93 ± 3.07 | 70.43 ± 3.13 | 71.39 ± 3.06 | 70.43 ± 3.13 | 661 | 20 |
| **Ensemble (DL Soft)** | **80.10 ± 0.35** | **78.45 ± 0.44** | **78.65 ± 0.43** | **78.45 ± 0.44** | **661** | **23** |
| CTT-MMC-A | 69.27 ± 2.87 | 65.37 ± 1.61 | 68.93 ± 2.09 | 75.09 ± 3.01 | 7200 | 60 |
| fastVideo | 68.21 ± 0.73 | 61.19 ± 0.53 | 65.86 ± 0.57 | 74.68 ± 0.68 | 10800 | 90 |
| TimeSformer | 64.98 ± 1.16 | 59.00 ± 1.07 | 63.26 ± 0.92 | 70.77 ± 0.94 | 43200 | 120 |
| DIViTA Swin-2D | 72.66 ± 1.37 | 67.68 ± 1.36 | 71.76 ± 1.09 | 77.49 ± 1.18 | 21600 | 100 |
| DIViTA Swin-3D | 75.57 ± 0.66 | 70.48 ± 0.41 | 74.21 ± 0.40 | 80.02 ± 0.47 | 28800 | 110 |

**Notes.**
Bold values highlight the best model results.

performance in more realistic, skewed distributions. sAP evaluates the model's precision at the instance level by first computing the average precision across all predicted labels for each sample and then averaging these scores across all instances. This metric provides insight into how well the model captures the correct combination of genres for individual movies. Together, these metrics offer a comprehensive evaluation, covering global, per-label, and per-instance performance while accounting for imbalanced genre distributions (*Zhang & Zhou, 2014*).

## Results on Trailers12K dataset

Table 3 compares various ML models (KNN, LR, NB, Random Forest (RF), SVM, and DT) and DL algorithms for multi-label genre classification using the Trailer12K dataset. As mentioned earlier, the Trailers12K dataset is divided into three data splits. The results are reported by averaging the metric values across these splits and the corresponding standard deviation for each average value along with training and testing times for all methods.

The performance of Transformer-based models is particularly noteworthy. BERT, RoBERTa, and DistilBERT performed better than traditional ML approaches, with µAP scores around 79–80%. This suggests that these models effectively capture the nuanced features and complex relationships between different genres. The soft ensemble of DL models achieves the best overall performance (µAP: 80.10, mAP: 78.45), indicating that combining predictions from multiple DL models can effectively capture different aspects of

genre classification. This approach benefits from the diverse strengths of individual models in recognizing various genre-specific features.

Among traditional ML methods, SVM performs well ($\mu$AP: 74.00 $\pm$ 0.04), significantly outperforming other classical algorithms. This suggests that SVMs are adept at handling the high-dimensional feature space. The performance gap between DL and traditional ML methods is substantial, with even the best traditional ML method, SVM, falling short of the worst-performing DL model AlBERT by about four percent in $\mu$AP. This underscores the effectiveness of DL in capturing complex, hierarchical features necessary for multi-label genre classification.

Compared with previous state-of-the-art methods, the proposed approach shows significant improvements. Our ensemble outperforms DIViTA Swin-3D (*Montalvo-Lezama, Montalvo-Lezama & Fuentes-Pineda, 2023*) by 4.53% in $\mu$AP and 7.97% in mAP. The advantage is even more pronounced against other video-based methods: 11.12% $\mu$AP over TimeSformer (*Bertasius, Wang & Torresani, 2021*), 7.89% over fastVideo (*Cascante-Bonilla et al., 2019*), and 5.31% over CTT-MMC-A (*Wehrmann & Barros, 2017*). These consistent gains across all metrics demonstrate the superiority of our text-based approach.

The relatively small standard deviations in most results indicate consistency across different data splits, suggesting robust performance. However, the higher variability in some ensemble methods, notably the hard ensemble of DL models, points to potential instability. The discrepancy between $\mu$AP and mAP scores across all models, with $\mu$AP consistently higher, suggests that the models perform better on more common genres and may struggle with rarer ones. This is a common challenge in multi-label classification and indicates an area for potential improvement in future research.

The computational performance reveals our DL Soft Ensemble achieves superior accuracy (80.10 $\mu$AP) with practical efficiency (661s training, 23ms testing), being 43$\times$ faster to train than video methods (DIViTA Swin-3D: 28800s) while outperforming them by 4.53 $\mu$AP.

### Results on LMTD9 test dataset

Table 4 compares various ML and DL algorithms' performance for multi-label genre classification using the LMTD9 test dataset. For these tests corresponding testing time is also reported in the table to reflect the efficiency, the training was performed on Trailers12K dataset.

In the context of traditional ML methods, SVM again demonstrates superior performance ($\mu$AP: 74.33, mAP: 71.13), consistent with the results observed on the Trailer12K dataset. LR is the second-best traditional ML method. The DL models, particularly the transformer-based architectures, significantly improve performance over traditional ML methods. BERT achieves the highest individual model performance ($\mu$AP: 80.16, mAP: 79.10), closely followed by RoBERTa and DistilBERT. This substantial improvement over traditional ML methods (about six percent in $\mu$AP compared to SVM) highlights the power of these language models in capturing nuanced features for genre classification.

**Table 4  Comparison of traditional ML and DL results with mean, standard deviation, and testing time on the LMTD-9 test set.**

| Model | μAP | mAP | wAP | sAP | Testing time (s) |
|---|---|---|---|---|---|
| Decision Trees | 64.24 ± 0.29 | 62.90 ± 0.32 | 60.79 ± 0.55 | 62.90 ± 0.32 | 1.02 |
| k-Nearest Neighbors | 66.56 ± 0.11 | 65.86 ± 0.12 | 65.35 ± 0.24 | 65.86 ± 0.12 | 1.69 |
| Logistic Regression | 71.35 ± 0.23 | 67.26 ± 0.07 | 68.83 ± 0.23 | 67.26 ± 0.07 | 1.01 |
| Naive Bayes | 70.00 ± 0.43 | 64.88 ± 0.53 | 66.95 ± 0.54 | 64.88 ± 0.53 | 1.01 |
| Random Forest | 63.89 ± 0.09 | 61.91 ± 0.20 | 61.86 ± 0.14 | 61.91 ± 0.20 | 1.13 |
| Support Vector Machine | 74.33 ± 0.14 | 71.13 ± 0.16 | 71.52 ± 0.33 | 71.13 ± 0.16 | 2.89 |
| Ensemble (ML Hard) | 71.81 ± 0.38 | 68.20 ± 0.43 | 69.22 ± 0.44 | 68.20 ± 0.43 | 6.80 |
| Ensemble (ML Soft) | 72.82 ± 0.39 | 69.21 ± 0.44 | 70.23 ± 0.45 | 69.21 ± 0.44 | 7.35 |
| AlBERT | 78.99 ± 0.36 | 76.86 ± 0.48 | 76.41 ± 0.43 | 76.86 ± 0.48 | 6 |
| BERT | 80.16 ± 0.35 | 79.10 ± 0.57 | 77.83 ± 0.58 | 79.10 ± 0.57 | 7 |
| DistilBERT | 79.45 ± 0.61 | 77.92 ± 0.64 | 76.81 ± 0.98 | 77.92 ± 0.64 | 5 |
| RoBERTa | 80.10 ± 0.41 | 79.10 ± 0.24 | 78.42 ± 0.38 | 79.10 ± 0.24 | 8 |
| Ensemble (DL Hard) | 73.70 ± 0.37 | 69.89 ± 0.42 | 71.60 ± 0.29 | 69.89 ± 0.42 | 12 |
| **Ensemble (DL Soft)** | **80.37 ± 0.33** | **79.39 ± 0.54** | **79.39 ± 0.45** | **79.39 ± 0.54** | **12** |
| VGG16+SVM, XGBoost | – | 73% | – | – | 21.9 |
| CTT-MMC-TN | 74% | 64% | 72% | – | 50.0 |
| C3DLSTM+VRFN | 74% | 64% | 72% | – | 70.3 |
| AFAnet+ASM | 75% | – | – | – | 90.4 |
| LOVA | – | 73% | 77% | – | 80.6 |

**Notes.**
Bold values highlight the best model results.

The soft ensemble of DL models yields the best overall results (μAP: 80.37, mAP: 79.39), marginally outperforming individual transformer models. This suggests that while ensemble methods still provide benefits, the gain is less pronounced than the Trailer12K dataset, possibly indicating that individual models are already performing near-optimally on this dataset.

Comparing these results with previous state-of-the-art methods reported in the literature, we see significant improvements. For instance, the best-performing soft ensemble outperforms the VGG16+SVM, XGBoost (*Mervitz et al., 2020*) approach by about six percent in mAP. It also shows substantial improvements over other reported methods like CTT-MMC-TN (*Wehrmann & Barros, 2017*), C3DLSTM+VRFN (*Bi, Jarnikov & Lukkien, 2021*), and LOVA (*Cai et al., 2023*) across various metrics. The performance gap between DL and traditional ML methods is even more pronounced on this dataset compared to Trailer12k. This suggests that the LMTD9 dataset has more complex or nuanced features that DL models are better equipped to capture. Interestingly, the hard ensemble of DL models performs poorly compared to individual models and the soft ensemble. This indicates that simple majority voting (hard ensemble) is ineffective for this dataset.

## Statistical significance analysis

We employed the Wilcoxon signed-rank test (*Taheri & Hesamian, 2013*) to compare the proposed ensemble model's performance against RoBERTa and DIViTA Swin-3D. This

**Table 5** Wilcoxon signed-rank test results comparing the proposed ensemble model with RoBERTa and DIViTA Swin-3D on Trailers12K and LMTD9 datasets.

| Metric | Dataset | Ensemble *vs.* RoBERTa (p) | Ensemble *vs.* DIViTA (p) |
|---|---|---|---|
| Micro-average Precision (μAP) | Trailers12K | 0.043 | 0.027 |
| | LMTD9 | 0.046 | 0.029 |
| Macro-average Precision (mAP) | Trailers12K | 0.038 | 0.021 |
| | LMTD9 | 0.040 | 0.024 |
| Weighted-average Precision (wAP) | Trailers12K | 0.041 | 0.030 |
| | LMTD9 | 0.044 | 0.028 |
| Sample-average Precision (sAP) | Trailers12K | 0.036 | 0.025 |
| | LMTD9 | 0.042 | 0.026 |

**Table 6** Average class-wise results of the proposed ensemble on the Trailers12K and LMTD-9 datasets.

| Classes | Trailer 12K | | | LMTD9 | | |
|---|---|---|---|---|---|---|
| | Precision | Recall | F1-score | Precision | Recall | F1-score |
| Action | 0.79 | 0.65 | 0.71 | 0.76 | 0.68 | 0.72 |
| Adventure | 0.72 | 0.50 | 0.59 | 0.83 | 0.59 | 0.69 |
| Comedy | 0.82 | 0.71 | 0.76 | 0.84 | 0.71 | 0.77 |
| Crime | 0.73 | 0.63 | 0.68 | 0.72 | 0.57 | 0.63 |
| Drama | 0.76 | 0.73 | 0.74 | 0.78 | 0.75 | 0.76 |
| Fantasy | 0.78 | 0.44 | 0.56 | – | – | – |
| Horror | 0.83 | 0.78 | 0.80 | 0.76 | 0.69 | 0.72 |
| Romance | 0.74 | 0.52 | 0.61 | 0.66 | 0.43 | 0.52 |
| Sci-fi | 0.81 | 0.65 | 0.72 | 0.81 | 0.66 | 0.73 |
| Thriller | 0.71 | 0.75 | 0.73 | 0.62 | 0.40 | 0.48 |

non-parametric test is ideal for paired samples and does not assume normal distribution, making it suitable for small or non-Gaussian data.

The test was applied across three experimental splits using four metrics: $\mu AP$, $mAP$, $wAP$, and $sAP$. A $p$-value below 0.05 indicates significant performance differences. As shown in Table 5, all $p$-values were below this threshold, confirming that the ensemble model significantly outperforms both baselines on the Trailers12K and LMTD9 datasets.

## Classwise results of ensemble

The results presented in Table 6 show the classwise results of the ensemble on the Trailer 12K and LMTD9 datasets. The ensemble demonstrates effective results, with F1-scores ranging from moderate to good across most genres. This suggests the ensemble approach is robust for this task. Notably, there's a consistent performance between the two datasets for most genres, indicating good generalization capabilities of the model.

Certain genres stand out as strong performers. Comedy and Drama consistently perform across both datasets, suggesting these genres possess distinctive features that the model can readily identify. Horror performs exceptionally well in Trailer 12K, though slightly less so in LMTD9. Conversely, genres like Adventure and Romance present challenges, with lower

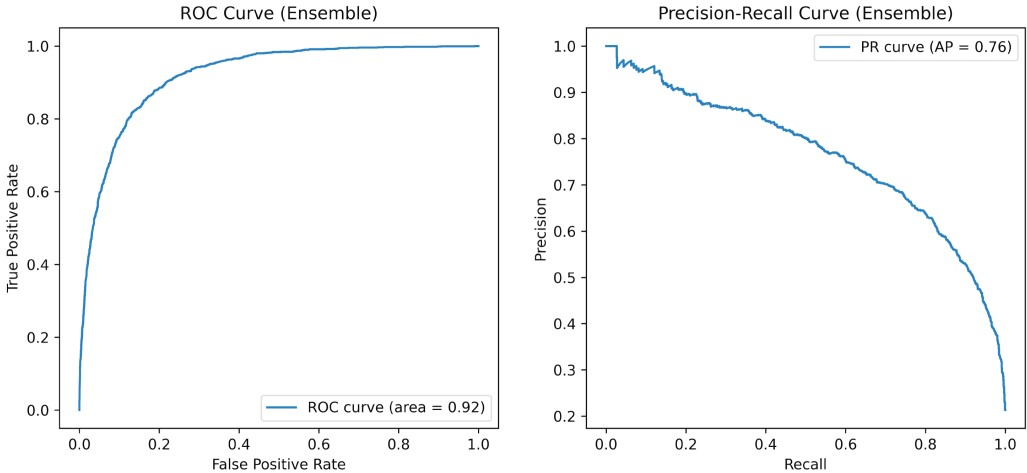

**Figure 2   ROC and PR Overall Curve of the proposed method on the Trailers12K dataset.**

F1-scores, especially in recall. This indicates difficulty in identifying all instances of these genres. Fantasy, present only in Trailer 12 K, shows a particularly low recall, suggesting the model struggles to identify many fantasy films.

Figure 2 illustrates the receiver operating characteristic (ROC) and precision-recall (PR) curves for the ensemble model on the Trailers12K dataset. The ROC curve, with an AUC of 0.92, demonstrates the model's ability to differentiate between classes across various thresholds. The steep rise near the top-left corner indicates excellent performance, particularly at strict thresholds. Meanwhile, the PR curve, featuring an average precision score of 0.76, reveals that the model effectively balances precision and recall. It starts with high precision and gradually decreases as recall increases, which is advantageous when minimizing false positives is crucial.

The performance drop for less frequent genres is further reflected in lower recall and F1-scores, indicating that class imbalance adversely affects the generalizability of the proposed ensemble method. However, by integrating diverse transformer-based models, the ensemble approach captures complementary patterns across genres, helping to mitigate this effect to some extent.

## Interpretability and error analysis using LIME

This section presents the results of local and global explainability analyses, providing insights into the model's genre predictions.

### Genre local explainability

Local explainability helps us understand how a model makes predictions for individual samples. Visualizing the importance of different words (tokens) in a movie plot synopsis reveals the extent to which each word contributes to predicting a specific genre. This insight enables us to better understand the model's decision-making process.

Figure 3 shows the local explanation for a plot correctly classified as "romance". The bar chart on the left-hand side presents the words that most strongly influenced the model's

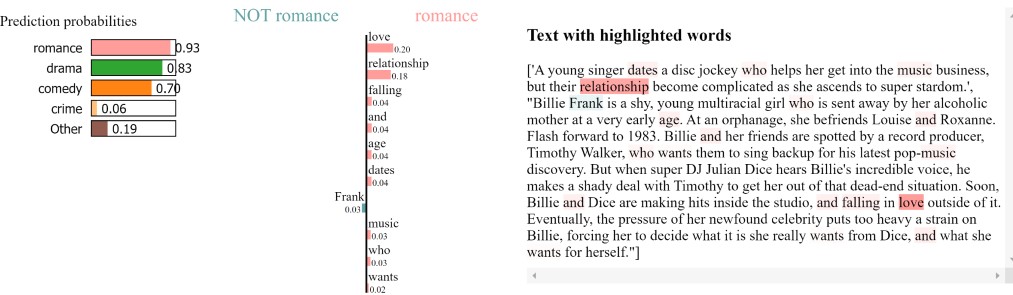

**Figure 3** LIME local explanation for the genre "Romance" in the plot of the movie Glitter (*Mark & Curtis-Hall, 2001*).

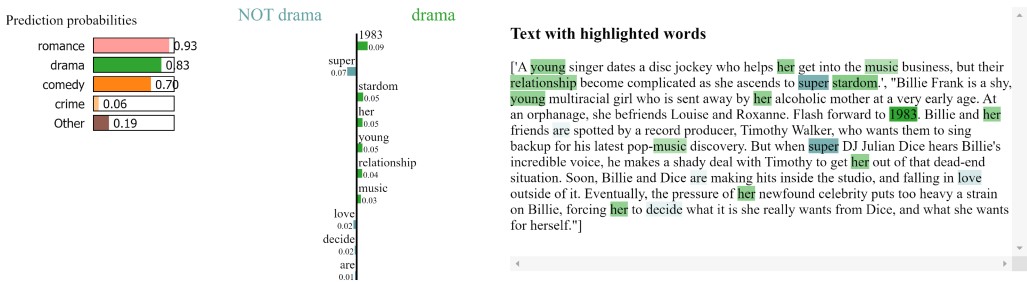

**Figure 4** LIME local explanation for the genre "Drama" in the plot of the movie Glitter (*Mark & Curtis-Hall, 2001*).

prediction, with the colour intensity indicating the strength of the influence. Words like "love," "relationship," and "falling" are highlighted as significant contributors to the "romance" classification. The right-hand side of Fig. 3 displays the corresponding text with these influential words highlighted within the context of the synopsis. Figure 4 follow a similar format, providing local explanations for synopses classified as "drama" respectively. In Fig. 4, words like "young," "super," and "stardom" are highlighted as key contributors to the "drama" prediction.

### *Genre global explainability*

Global explainability was conducted to better understand the model's behaviour. This involves identifying the most important words (tokens) for each genre (class) in the dataset.This is achieved by aggregating the word importance values derived from LIME's local explanations across multiple samples within each genre and calculating the average importance score for each word.

Figures 5A–5B presents the top 10 most relevant words and their corresponding probabilities for each genre. For instance, in Fig. 5A, words like "passion," "kiss," and "affection" exhibit high probabilities for the "romance" genre, while in Fig. 5B, words like "conflict," "emotion," and "struggle" have high probabilities for the "drama" genre. This analysis reveals distinctive patterns and word-class associations, therefore proving necessary for handling a thorough performance assessment and error analysis. By identifying the

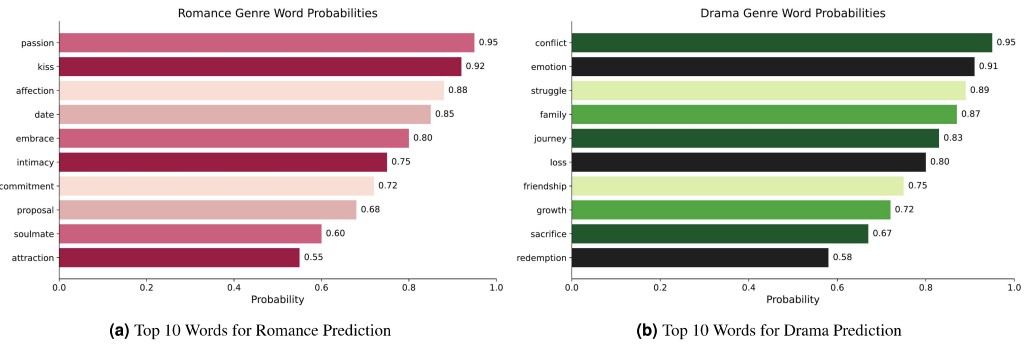

**(a)** Top 10 Words for Romance Prediction   **(b)** Top 10 Words for Drama Prediction

**Figure 5** **(A–B) LIME global explanations for different genres with word probabilities on the Trailers12K dataset.**

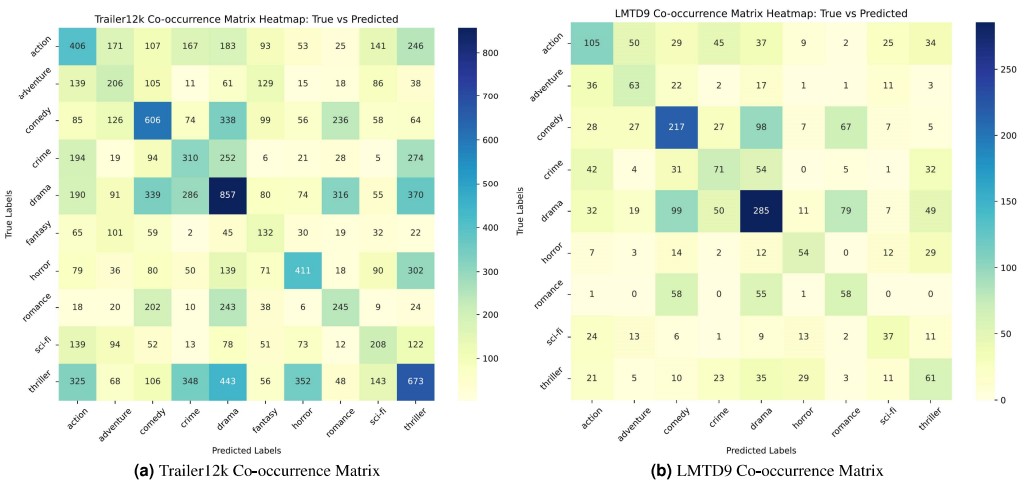

**(a)** Trailer12k Co-occurrence Matrix   **(b)** LMTD9 Co-occurrence Matrix

**Figure 6** **(A–B) Co-occurrence matrix heatmaps for the Trailer12k and LMTD9 datasets.**

most dominant words for each genre, we can more effectively understand the model's classification results and potential areas for improvement.

## Co-occurence matrix

A co-occurrence heatmap is utilized to evaluate the model's performance in the multi-label, multi-class setting. Figures 6A and 6B show the co-occurrence heatmaps for the LMTD9 and Trailer12k datasets for the proposed ensemble method. The heatmaps display the model's predictions, with true labels on the vertical axis and predicted labels on the horizontal axis. Matching cells indicate correct predictions, while off-diagonal cells highlight errors.

From the heat maps, varying performance between different genres can be observed.For instance, in Fig. 6A, the model exhibits lower accuracy for genres like "sci-fi" and "thriller,". Similarly, in Fig. 6B, genres like "adventure" and "fantasy" show relatively lower accuracy compared to others. This performance variation can be attributed to several factors. Trailer12k and LMTD9 datasets have an imbalanced distribution of samples across different

genres. Additionally, the semantic and thematic similarities between specific genres, such as "sci-fi" and "action," or "fantasy" and "adventure," can make it challenging for the model to differentiate between them accurately.

## CONCLUSIONS

This study introduces a novel ensemble deep learning model for multi-label movie genre classification, using textual movie plots. By combining BERT, RoBERTa, and DistilBERT, our model captures nuanced genre-specific information, achieving state-of-the-art performance on benchmark datasets and surpassing existing approaches. We also integrate LIME to provide local and global explanations for the model's predictions, enhancing interpretability and shedding light on the decision-making process. While the model performs well overall, class imbalance remains a challenge, particularly because movie plots often belong to multiple genres, leading to uneven label distribution. Despite this, our ensemble of transformer-based models and balanced training strategies helps improve predictive accuracy. The proposed ensemble model can be easily integrated into existing recommendation systems or streaming platforms. Using only plot data enables lightweight deployment without heavy video or audio processing. It can serve as a backend module for genre tagging, improving personalized content delivery.

However, the study has certain limitations, including the datasets, an emphasis mostly on English-language movies and showing class imbalance, confining generalizability to non-English or understated genres. The dependence on textual data alone may ignore genre-specific keys from multimodal sources like audio and visuals. While LIME improves interpretability, its reliance on perturbed inputs can affect reliability, particularly in complex multi-label tasks. SHAP or attention-based models can be explored to enhance trust in genre predictions. Additionally, since the model is trained on a fixed set of predefined genres, it may resist adapting to emerging or evolving genres. While efficient compared to alternatives, the ensemble's GPU requirements may challenge mobile deployments.

Future research aims to incorporate additional textual features like movie scripts, subtitles, and reviews to further improve accuracy and understanding of genre characteristics. Expanding the model to include social media discussions could also reveal public sentiment and genre trends. The proposed model has implications for enhancing movie recommendation systems, content-based retrieval, and predictive analysis. Further research avenues include cross-lingual genre classification, analysis of genre evolution, and multimodal analysis integrating visual and auditory features. Additional avenues for research include cross-lingual genre classification, analysis of genre evolution, and multimodal analysis that integrates visual and auditory features.

### Funding

This work was supported by Taif University, Saudi Arabia, Researchers Supporting Project number (TU-DSPP-2024-41), Taif University, Saudi Arabia. The funders had no role

in study design, data collection and analysis, decision to publish, or preparation of the manuscript.

## Grant Disclosures

The following grant information was disclosed by the authors:
Taif University, Saudi Arabia, Researchers Taif University, Saudi Arabia: TU-DSPP-2024-41.

## Competing Interests

The authors declare there are no competing interests.

## Author Contributions

- Faheem Shaukat conceived and designed the experiments, performed the computation work, prepared figures and/or tables, and approved the final draft.
- Naveed Ejaz conceived and designed the experiments, performed the experiments, analyzed the data, performed the computation work, prepared figures and/or tables, and approved the final draft.
- Zeeshan Ashraf analyzed the data, performed the computation work, authored or reviewed drafts of the article, and approved the final draft.
- Mrim M. Alnfiai conceived and designed the experiments, prepared figures and/or tables, funding, and approved the final draft.
- Nouf Nawar Alotaibi performed the experiments, analyzed the data, authored or reviewed drafts of the article, and approved the final draft.
- Salma Mohsen M. Alnefaie analyzed the data, authored or reviewed drafts of the article, and approved the final draft.

## Data Availability

The data and code are available at GitHub and Zenodo:

- https://github.com/faheemshaukat/LIME

- faheemshaukat. (2025). faheemshaukat/LIME: Movie Genre Classification Using Ensemble Learning & LIME –v1.0 (v1.0). Zenodo. https://doi.org/10.5281/zenodo.14906135.

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
