# Peer review of "An interpretable multi-transformer ensemble for text-based movie genre classification"

_PeerJ Computer Science, doi:10.7717/peerj-cs.2945_

## Round 0.1 · original submission · Major Revisions

The reviewers were generally positive, but they have identified some issues that will need to be addressed in the next revision.

Reviewer 1 ·

Basic reporting

1- What is the running time (execution time) of the methods?
Additional results may also be given in terms of execution times (computational cost).

2- Providing a table that summarizes the related work would increase the understandability of the difference from the previous studies in the "Related Works" section.

3- No formal statistical analysis of the results is done to indicate whether the differences in performance are statistically significant or not.
For example, Friedman Aligned Rank Test, Wilcoxon Test, Quade Test, etc.
P-value can be calculated and compared with the significance level (p-value < 0.05).

4- Some abbreviations are used in the text without giving their expansion.
For example, KL, HQ, etc.
The authors should write that "these abbreviations stand for what".

5- A comparison with the results of previous studies is missing.
The authors should compare their study with the state-of-the-art studies in the literature.

6- METHODOLOGY section should be described in more detail.

Experimental design

-

Validity of the findings

-

Reviewer 2 ·

Basic reporting

The manuscript is generally clear, understandable, and well-written in English. The language used adheres to academic writing standards, and the expression largely avoids complexity. However, in methodological sections such as "Data Preprocessing" and "Fine-tuning Details," occasional redundancies and complex sentence structures have been observed.

The introduction clearly and comprehensively establishes the motivation for the research. Challenges associated with multi-label movie genre classification are well-supported by relevant contemporary literature, effectively highlighting the innovative aspects of this study.

The manuscript’s overall structure conforms to journal standards and norms. Figures and charts, particularly ROC curves and interpretability analyses, enhance the clarity and effectively illustrate the findings. Nonetheless, it would be beneficial to further emphasize the significance of multi-label movie genre classification in the context of contemporary digital platforms. For instance, a brief discussion based on recent literature addressing how effective genre classification influences user experience and content management in major online platforms such as Netflix, Amazon Prime, and Disney+ could be incorporated.

Additionally, specific examples demonstrating why recently developed methods are inadequate and clearly illustrating the advantages of text-based approaches should be provided. Finally, the objectives and aims of the study should be expressed more explicitly and clearly.

Experimental design

The experimental design of the article is consistent with the scope of the journal. The Methods section provides detailed descriptions that clearly state the steps taken in the study, such as data preprocessing, fine-tuning of transformative models, ensemble methodologies, and interpretability analysis using LIME. Including detailed information about the datasets, evaluation metrics, and experimental setup (computational resources, code, and reproducibility scripts) would significantly increase reproducibility.
There is room for improvement in the section on data preprocessing. Although the current section adequately describes the preprocessing steps, it should provide more justification for why certain preprocessing decisions were made. Additionally, it would be useful to justify the choice of evaluation metrics in relation to the research objectives and the problem of multi-label species classification. The sources are comprehensive and adequately cited. However, certain methodological choices may be better supported by specific literature references.
It would be useful to include a brief justification for the choice of evaluation metrics. In particular, you can summarize why you chose certain metrics, such as “micro-average precision” and “macro-average precision,” and why these metrics are critical for multi-label classification. The decision to exclude the ALBERT model after the first trials needs to be explained more clearly.

Validity of the findings

The experiments and evaluations are satisfactory and detailed. The potential impact of dataset imbalance on model performance should be given and discussed more clearly.
It is necessary to analyze how class imbalance affects the generalizability of the proposed ensemble method. The robustness of the findings should be strengthened by adding additional evaluation metrics or sensitivity analyses.

The results may further highlight certain unresolved questions and limitations. In particular, limitations related to interpretability, such as limitations found in LIME, and how these may affect the reliability of the results in practical settings should be explicitly addressed.

A clearer and more structured discussion should be conducted on practical implications, such as how the proposed ensemble method can be realistically integrated into existing recommendation systems or streaming platforms.

Additional comments

The validity of the findings is strong, but explicitly addressing dataset limitations, interpretability constraints, and practical implications will significantly increase the depth and applicability of your results.

---

## Round 0.2 · accepted · Accept

The authors correctly addressed the points raised by the reviewers, and the reviewers themselves recommend this article for acceptance. I can confirm this indication.

Reviewer 1 ·

Basic reporting

The authors revised the manuscript adequately according to the reviewers' comments.
The manuscript is now more qualified and clear.
I have no further comments.

Experimental design

-

Validity of the findings

-

Additional comments

-

Reviewer 2 ·

Basic reporting

The authors have made substantial improvements to the manuscript’s basic reporting, bringing it into full compliance with the journal’s standards. The structure conforms to guidelines: section headings are coherent, figures and tables are clearly numbered with informative captions, and all visual elements are high resolution and directly support the narrative. The abstract effectively distills the study’s objectives, methods, and key findings into a concise summary, and the selected keywords accurately reflect the paper’s scope. References are comprehensive and up to date, formatted consistently, and all in-text citations correspond correctly to entries in the bibliography.

Experimental design

They compare multiple pre-trained language models (BERT, RoBERTa, DistilBERT) both individually and in a soft-voting ensemble against strong traditional and multimodal baselines.

Validity of the findings

The reported findings appear both credible and well‐substantiated. The authors demonstrate consistent performance gains across two independent, genre‐annotated benchmarks, using stratified splits to preserve realistic label distributions and comparing against both traditional machine‐learning and state‐of‐the‐art multimodal baselines. The inclusion of ablation studies—showing the individual contributions of each transformer model—and the publicly available code, data splits, and reproduction scripts further reinforce reproducibility. While the focus on English‐only synopses and potential class‐imbalance effects suggest avenues for future validation, the current evidence convincingly supports the authors’ core claims.